# Development of Freeze-Thaw Stable Starch through Enzymatic Modification

**DOI:** 10.3390/foods10102269

**Published:** 2021-09-25

**Authors:** Seung-Hye Woo, Ji-Soo Kim, Hyun-Mo Jeong, Yu-Jeong Shin, Jung-Sun Hong, Hee-Don Choi, Jae-Hoon Shim

**Affiliations:** 1Department of Food Science and Nutrition, The Korean Institute of Nutrition, Hallym University, Hallymdaehak-gil 1, Chuncheon 24252, Korea; shye94@hallym.ac.kr (S.-H.W.); Jisoo1226@hallym.ac.kr (J.-S.K.); 41351@hallym.ac.kr (H.-M.J.); yujeong0115@hallym.ac.kr (Y.-J.S.); 2Division of Strategic Food Research, Korea Food Research Institute, Nongsaengmyeong-ro 245, Iseo-myeon, Wanju-gun 55365, Korea; jungsunhong@kfri.re.kr (J.-S.H.); chdon@kfri.re.kr (H.-D.C.)

**Keywords:** freeze-thaw stability, amylopectin cluster, clean label starch, branching enzyme, cyclodextrin glucanotransferase

## Abstract

The use of unmodified starch in frozen foods can cause extremely undesirable textural changes after the freeze-thaw process. In this study, using cyclodextrin glucanotransferase (CGTase) and branching enzymes, an amylopectin cluster with high freeze-thaw stability was produced, and was named CBAC. It was found to have a water solubility seven times higher, and a molecular weight 77 times lower, than corn starch. According to the results of a differential scanning calorimetry (DSC) analysis, dough containing 5% CBAC lost 19% less water than a control dough after three freeze-thaw cycles. During storage for 7 days at 4 °C, bread produced using CBAC-treated dough exhibited a 14% smaller retrogradation peak and 37% less hardness than a control dough, suggesting that CBAC could be a potential candidate for clean label starch, providing high-level food stability under repeated freeze-thaw conditions.

## 1. Introduction

Bread is one of the most widely consumed foods in the world and baking is one of the oldest known cooking technologies [1,2,3]. Novel ingredients and equipment have been continuously introduced to develop higher-quality bread, and research on baking has shown steady and impressive progress over many years [2]. In particular, the frozen dough method has significantly reduced the labor requirements and cost of baking [4,5,6]. However, bread prepared from frozen dough tends to be of lower quality and becomes stale more rapidly than bread prepared from fresh dough [5,7,8]. Therefore, the frozen dough method is not generally used for making white bread, but is typically used for making sweet bread prepared with additives such as oils, sugars, and sugar alcohols [8]. The baking industry has increasingly adopted freezing technology due to the economic benefits of the centralized manufacturing and distribution process, as well as consumer demand for standardized product quality [2,9,10]. Numerous studies have been conducted to determine the optimal storage conditions for frozen dough to obtain high-quality baking products [11,12,13,14].

Dough can be damaged during frozen storage due to ice crystal formation [8,15,16]. In addition to this damage, ice crystal formation also has a negative effect on yeast stability [8]. Some of the freezable water that does not bind to gluten during dough formation is frozen when the dough is stored during frozen storage [9,17,18]. The ice crystals formed during frozen storage may cause physical damage to the gluten protein structure, resulting in weak hydrophobic bonds, redistribution of water in the dough gluten network, and a loss of gas-holding capacity during baking. Repeated freeze-thaw cycles during frozen storage exacerbate such phenomena [9,10]. Therefore, harsh processing conditions, such as freeze-thaw cycles, can result in substantial deterioration of the dough structure [9]. In addition, the size of the ice crystals may increase in proportion to the extent of frozen storage, which reduces yeast viability in the frozen dough [8,17].

Various additives have been suggested to improve the quality of frozen dough [4,13,15,19,20]. β-glucan is able to bind water molecules released from the gluten-starch network and inhibit the growth of ice crystals [4]. Because soy protein binds tightly to water molecules, it can be added to reduce the damage caused by the freezing of dough [5]. The addition of xanthan gum or hydroxypropyl methylcellulose (HPMC) to frozen dough has also been reported to improve product volume and quality during long-term frozen storage [15,19]. However, the use of these additives in the actual food industry can be difficult due to consumer concerns about food safety [10,21].

Starch is commonly used in various food products [22]. However, the use of native starches in the food industry is limited by various inherent problems, such as an inability to tolerate high shear stress, high retrogradation, poor water solubility, and low freeze-thaw stability [23,24,25]. Therefore, to improve its functionality and stability, starch has been chemically modified in industrial applications [22,25]. However, chemical modification is viewed as undesirable by consumers because chemical residues may remain in the final product [22,23,26]. Recently, consumer concerns regarding chemically modified starch have led to a preference for ‘clean label’ modified starches [24,26]. These modified starches are prepared using naturally occurring materials and hydrothermal and enzymatic processes, without any need for synthetic chemicals [22,24,25]. For example, stearic acid and hydrothermal treatments have replaced chemical processes in the production of chemically cross-linked starch [22]. A natural functional ingredient derived from flax seed is now being used as a replacement for chemical dough conditioner [27]. In addition, to adjust the digestibility of starch, various enzymatic clean label modified starches have been developed [28,29]. Although various studies have been conducted to investigate the development of modified starch, with specific functional properties caused by chemical, physical, or enzymatic modifications [30], there have been few studies on clean label freeze-thaw-resistant starch.

In this study, we hypothesized that the form of modified starch with many short branches could represent a material with the potential to reduce the release of water molecules from dough, thus helping to maintain the quality of the dough during freeze-thaw storage. Therefore, we prepared an enzymatically modified dough and investigated its applicability to bread baking.

## 2. Materials and Methods

### 2.1. Enzymatic Modification of Starch

A corn starch suspension (1%, *w*/*v*) was prepared in 50 mM sodium acetate buffer (pH 6.0, 50 mL). After preheating at 60 °C for 5 min, the starch solution was incubated while stirring with cyclodextrin glucanotransferase (CGTase, 0.645 KNU-CP/mg substrate) and the branching enzyme (0.000001872 BEU/mg substrate) at 60 °C for a specific time, to produce CGTase and branching enzyme-treated corn starch (CBAC). Cyclodextrin glucanotransferase (CGTase; Toruzyme) and a branching enzyme (Branzyme) were purchased from Novozymes (Bagsvaerd, Denmark) and Isoamylase derived from *Pseudomonas* sp. was purchased from Megazyme (The Bray Co., Wicklow, Ireland). One KNU-CP of CGTase was defined as the amount of enzyme that breaks down 5.26 g starch per hour according to Novozyme’s standard method for determining alpha-amylase. One BEU was defined as the quantity of enzyme that causes a decrease in absorbance at 660 nm of an amylose-iodine complex of 1% per minute under standard conditions (pH 7.2, 60 °C). After the enzyme reaction, enzyme mixtures were boiled for 20 min to halt enzyme activity.

The liquid form of CBAC starch prepared using the above method was lyophilized to attain a powder form. First, the oligosaccharides produced from the starch during the enzymatic treatment were removed using ethanol. The CBAC was precipitated using two volumes of ethanol at −25 °C and collected by centrifugation (5500× *g* at 4 °C for 40 min). The precipitates were suspended in distilled water with the same relative ratio as the CBAC starch solution, and then boiled for 5 min to ensure complete dissolution. The samples were frozen at −40 °C before freeze-drying using an Operon instrument (Gimpo, Korea).

### 2.2. Molecular Weight Distribution of Enzyme-Treated Starches

Starch suspension (2.5%, *w*/*v*) was prepared using distilled water. The starch was left in distilled water for 30 min to hydrate sufficiently. Dimethyl sulfoxide (DMSO) was then added to the starch suspension to achieve a 1% (*w*/*v*) concentration. After boiling for 1 h at 100 °C, the mixture was stirred overnight at room temperature. The sample and ethanol were mixed at a ratio of 1:6 and the mixture was centrifuged to remove the supernatant. The precipitate was dispersed in boiling water and boiled for 30 min with continuous stirring. Finally, the starch solution (2 mg/mL) was filtered for analysis using a 5-μm syringe filter (Acrodisc 25 mm; Pall Co., Port Washington, NY, USA). The weight-average molecular weight (M_w_) and M_w_ distribution curves were determined using a size exclusion chromatography–multiangle laser light scattering-refractive index detector (SEC-MALLS-RI). The SEC separation was performed with a high-performance size exclusion chromatograph (Agilent 1100; Agilent Technologies, Inc., Santa Clara, CA, USA) including a degasser, autosampler, pump (Waters 510; Waters Co., Milford, MA, USA), guard column (TSK PWH; Tosoh Co., Tokyo, Japan), and SEC column (Shodex SB-804 HQ, SB-806 HQ OHpak; Showa Denko, Tokyo, Japan), which were connected to a MALLS detector (Dawn DSP; Wyatt Technology, Goleta, CA, USA) and a refractive index detector (Waters 410; Waters Co.). The column was kept at 55 °C. The flow rate of the mobile phase (water) was 0.6 mL/min. The calculation of M_w_ was performed using Astra 472 software (Wyatt Technology) with the Berry extrapolation method and a dn/dc value of 0.185 mL/g.

### 2.3. Side Chain Length Distribution Analysis

The side chain length distribution of starch samples was determined using a method that we described previously [31]. Before the analysis, the sample was pretreated to facilitate the structural analysis of the starch. Corn starch and enzymatically modified starch was dissolved in 50 mM sodium acetate (pH 4.5) and was reacted with isoamylase at 40 °C for 96 h. The reaction solution was boiled for 5 min to stop the enzyme reaction. Then, pretreated starch was analyzed using high-performance anion exchange chromatography (HPAEC). The HPAEC system consisted of a CarboPac PA1 guard column (4 × 50 mm; Dionex, Sunnyvale, CA, USA), a CarboPac PA1 column (4 × 250 mm; Dionex), and a pulsed amperometric detector (ED40; Dionex). The samples (20 µL) were injected into the column and eluted with multiple gradients of 600 mM sodium acetate in 150 mM NaOH at a flow rate of 1 mL/min. The linear gradients of sodium acetate were as follows: 10−30% for 0−10 min, 30−40% for 10−16 min, 40−50% for 16−27 min, 50−60% for 27−44 min, 60−65% for 44−63 min, 65−66% for 63−70 min, and 66−100% for 70−71 min.

### 2.4. Determination of Starch Water Solubility

The CBAC powder was mixed with distilled water and supersaturated. The suspensions were boiled for 5 min at 100 °C. The samples were then cooled to 20 °C and the supernatant was collected via centrifugation at 11,200× *g* for 20 min. The modified starch water solubility was obtained by mixing the modified starch solutions and iodine solution at a ratio of 1:1, and reacting the mixture at 20 °C for 5 min. The measurements were determined using a spectrophotometer (Multiskan FC; Thermo Scientific, Waltham, MA, USA) operated at a wavelength of 550 nm. The standard curve covered a concentration range of 0.1%–0.5% (*w*/*v*). The results are expressed as the mean and standard deviation (SD) of three measurements per sample.

### 2.5. Preparation of Frozen Dough and Bread

The dough was prepared for a white bread mix (CheilJedang, Seoul, Korea) and a bread maker (Cuchen, Seoul, Korea) was used. The control dough was prepared using only 376 g of wheat flour, 4 g of yeast, and 220 mL of water. The CLS (Commercial clean label starch, Novation Proma 300; Ingredion, Westchester, Ireland) group and CBAC group doughs were treated with modified starches to achieve a CLS concentration of 10% (*w*/*w*) and a CBAC concentration of 5% (*w*/*w*), respectively. After that, the ingredients for the dough were kneaded for 30 min and the first fermentation proceeded for 120 min at 30 °C. After the first fermentation, the dough was sealed inside a polyethylene bag, frozen-stored for 22 h at −25 °C, and thawed for 2 h at 30 °C. This cycle was repeated three times. After the three freeze-thaw cycles, the dough was fermented again for 100 min at 30 °C and the bread was baked for 40 min at 190 °C. The bread was left to cool for 3 h and then sealed with a polyethylene bag and stored in a refrigerator at 4 °C.

### 2.6. Bread Loaf Volume

The bread loaf volumes were obtained after they had cooled for 3 h. The volume of each loaf was determined via the rice grain method [32]. The bread was placed in a 4 L container and the empty space was filled with rice grains. The bread volume was determined by measuring the volume of rice grains displaced from the 4 L container.

### 2.7. Texture Profile Analysis (TPA)

The TPA was performed with a texture analyzer (TMS-Pro; Food Technology Co., Sterling, VA, USA). The changes in the textural properties of each bread loaf were analyzed after 7 days of storage at 4 °C. The breads were measured by cutting four 20-mm-thick slices. Twice-repeated compression tests were conducted on the bread slices using a texture profile analyzer equipped with a 25 N load cell. The bread slices were compressed to a thickness of 12 mm (60%) with a 50-mm probe at a speed of 60 mm/min. The crumb firmness and elastic recovery were calculated based on the force-distance curves generated by two cycles of compression. The results are expressed as the mean and SD values of three measurements per bread slice.

### 2.8. Differential Scanning Calorimetry (DSC) Analysis

Bread staling was determined using a DSC 214 Polyma system (Netzsch, Selb, Germany) after 7 days of storage at 4 °C. The DSC was evaluated by making measurements directly on breadcrumbs without further pretreatment. Breadcrumbs were obtained from the center of the loaves after the aging process. The bread samples (10 mg) were weighed and sealed in aluminum pans. An empty pan was used as a reference. The pans were heated from 20 to 90 °C at a heating rate of 5 °C/min. Retrogradation was determined as the enthalpy, calculated based on the area under the endothermic peak located between 40 and 70 °C.

The freezable water in the dough was monitored with the Polyma DSC system using a previously reported method [4,9]. Five milligrams of the thawed dough was extracted from the center of each sample and placed directly into an aluminum pan, which was then sealed. The sample pan was analyzed alongside an empty crucible for reference. The analysis process involved two temperature control loops. In the first step, the samples were cooled from their initial temperature to −50 °C and held isothermally for 5 min. The sample temperature was then elevated from −50 to 20 °C at a rate of 5 °C/min. The enthalpy of the freezable water was determined as the energy absorbed during the melting of ice in the frozen dough.

### 2.9. Statistical Analysis

All measurements were repeated at least three times for each sample. The Kruskal–Wallis H test was used to test for significant differences among samples. All analyses were performed using SPSS software (version 25.0; SPSS Inc., Chicago, IL, USA) with *p* < 0.05 taken to indicate statistical significance.

## 3. Results

### 3.1. Structural Analysis of Enzyme-Modified Starch

The structural characteristics of freeze-thaw stable starch were analyzed via HPAEC. Compared to the corn starch, in CBAC the relative amounts of side chains below DP 10 and in the range of DP 28 to DP 47 increased. Consequently, the relative amounts of the other chain lengths in CBAC (DP 10–27, >DP 48) decreased (Figure 1). Both CBAC and CLS typically contained larger amounts of B2 and B3 chains than corn starch. In particular, CBAC possessed relatively more long side chains (DP 31–47) than CLS (Figure 1).

### 3.2. Solubility of Modified Starch

In Table 1, the changes in water solubility of enzyme-modified starches are shown by enzyme reaction time. The water solubility of corn starch was 12.4 mg/mL, whereas that of CBAC-3 h was 97.6 mg/mL, i.e., approximately seven times higher than that of the normal corn starch. However, an enzyme treatment of longer than 3 h gradually decreased the CBAC solubility.

### 3.3. Molecular Weight Distribution of Enzyme-Treated Starch

The M_w_ of starch was determined according to the enzyme reaction time using the SEC-MALLS-RI system. The M_w_ of corn starch before the enzyme reaction was 1.62 × 10^8^ Da. After the CGTase and branching enzyme treatment, the M_w_ of starch decreased to 2.09 × 10^6^ Da. After a 1-h reaction, the M_w_ remained constant (Figure 2). Interestingly, despite the rapid decrease in M_w_, there was virtually no production of small-molecule sugars, suggesting that the hydrolysis reaction of the enzymes acted on the amylopectin cluster units.

### 3.4. Effects of CBAC on Bread Baking

To investigate the effect of modified starch on freeze-thaw stability, CLS and CBAC were added to bread dough, respectively. The fermented doughs were subjected to the freeze-thaw process three times, as described in the Material and Methods, and the bread was then baked. As shown in Figure 3, the bread samples were compared in terms of volume; the volumes of the control, CLS-, and CBAC-added breads were 1833.33, 1706.67, and 1993.33 cm^3^, respectively. Relatively, the bread made with the CBAC had the largest volume (9% larger than the control bread volume). However, there was no significant difference in volume between the bread made with CLS and the control bread.

In the same bread loaves, textural changes were measured using TPA after storage for 7 days at 4 °C. The differences in the bread samples due to the type of starch used are shown in Table 2. The control bread was 37% harder than the bread baked with CBAC. In addition, compared to CLS, the CBAC was more effective in maintaining the bread crumb softness. There was no significant difference in springiness all among the three kinds of bread. Therefore, CBAC was considered to be effective for increasing bread crumb softness.

The retrogradation rate of the bread was also analyzed using DSC (Figure 4). The degree of retrogradation is usually expressed in terms of the enthalpy from 45 to 70 °C. Figure 4 shows the retrogradation of the breads treated with different starches after storage. The retrogradation peak of the breadcrumbs was 9.01 mJ for the control bread, 7.74 mJ for the bread treated with CBAC, and 8.68 mJ for the bread treated with CLS. Thus, among the bread samples, the bread made with the CBAC had the smallest retrogradation peak (14% smaller than that of the control bread). The use of CLS resulted in a retrogradation peak that was only 3% smaller than that of the control bread.

### 3.5. Freezable Water in Frozen Doughs

DSC can be used to study the influence of freeze-thaw treatments on the thermodynamic properties of dough and bread [5,7,9,15]. Figure 5 shows the freezable water in the frozen dough with the addition of different modified starches during storage. Before the freeze-thaw process, there was no significant difference in freezable water enthalpy among doughs prepared with different modified starches. However, after the third cycle of the freeze-thaw process, the dough made with CBAC was significantly different to the control dough in terms the peak enthalpy of freezable water. The freezable water enthalpy values of the control dough and dough made with the addition of CBAC were 122.47 and 98.64 mJ, respectively. This indicated that CBAC was able to effectively retain water molecules in dough during a freeze-thaw cycle.

## 4. Discussion

To investigate the starch structure of enzyme-modified starch, we analyzed the side chain length distribution and M_w_ distributions of modified starches (Figure 1 and Figure 2). As shown in Figure 2, the average M_w_ of the reaction products decreased sharply from 1.62 × 10^8^ to 2.09 × 10^6^ Da, indicating that amylopectin in starch was degraded into cluster units via cleavage of the inter-chain part between clusters. In the comparison of the side chain length distributions, CBAC starch had larger amounts of short chains below DP 10 and side chains ranging from DP 28–47 than the other starches (Figure 1). As shown in Table 1, CBAC had a higher water solubility than the original corn starch, which is consistent with several previous studies [33,34]. Usually, amylopectin clusters with a low M_w_ have higher water solubility than branched amylose [35]. The highly branched amylopectin clusters produced by the enzyme modification in this study had a higher water solubility than reported previously for amylopectin clusters [35]. Thus, the high water solubility of CBAC in our study could be due to the increase in the amount of short branches (<DP 10) and decrease in the M_w_ (Table 1 and Figure 1). The increase in the amount of chains in the range of DP 28–47 was likely caused by hydrolysis of the inter-chain region between amylopectin clusters. In addition, due to the branching enzymatic reaction, many short branches (<DP 10) were formed in amylopectin clusters. It has been reported that the more short branches a starch has, the more effectively it prevents syneresis [36]. In general, starches with a higher amylopectin content are known to exhibit higher freeze-thaw stability, because the numerous amylopectin branches prevent the separation of water molecules in the gel network [36,37,38].

Generally, bread baked with frozen dough is undesirable in terms of its textural and staling properties [39]. In this study, doughs mixed with enzymatically modified starches were exposed to a freeze-thaw process to evaluate their freeze-thaw stability, with the changes in their textural and staling properties then analyzed (Table 2, Figure 4). Despite repeated the freeze-thawing of the frozen dough, the addition of CBAC maintained the bread volume, and resulted in lower levels of hardness and bread staling relative to a control bread, indicating that CBAC improved the water holding capacity of starch under freeze-thaw conditions. The higher the water holding capacity, the lower the hardness of the bread [15]. This result was consistent with the results of other studies [40,41] indicating that the retrogradation of bread is positively correlated with the hardening of breadcrumbs.

## 5. Conclusions

In this study, we developed a simple method for preparing enzymatically modified starch (CBAC) with cyclodextrin glucanotransferase (CGTase) and branching enzymes. It showed a water solubility that was seven times higher and a M_w_ that was 77 times lower than corn starch. The addition of CBAC maintained the characteristics of dough during a freeze-thaw process, whereas the addition of CBAC to the frozen dough resulted in an antistaling effect during storage of the resulting bread. Therefore, CBAC could be considered a beneficial agent for frozen ready meals, and should have a role in the Asian frozen food market, which is growing dramatically [42].

## Figures and Tables

**Figure 1 foods-10-02269-f001:**
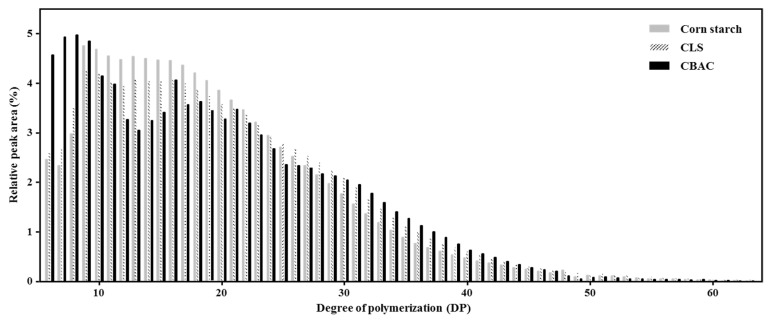
Properties of the modified starch structures.

**Figure 2 foods-10-02269-f002:**
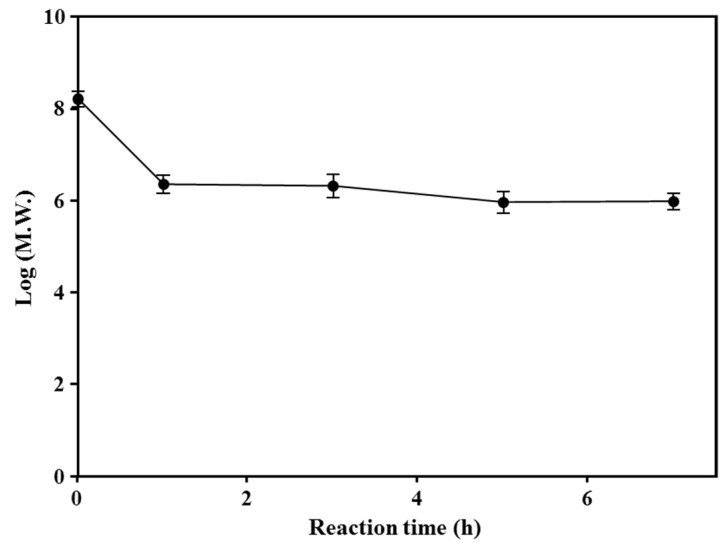
Time course analysis of the molecular weight (M_w_; g/mol) of starch during enzymatic modification. The Mw of corn starch samples treated with (CGTase) and a branching enzyme with various incubation times was measured using size exclusion chromatography-multiangle laser light scattering (SEC-MALLS).

**Figure 3 foods-10-02269-f003:**
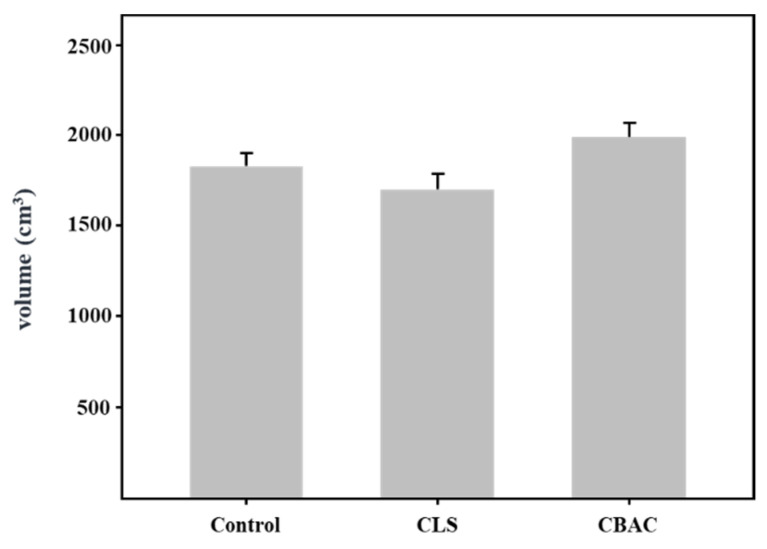
Volumes of the bread loaves treated with different starches.

**Figure 4 foods-10-02269-f004:**
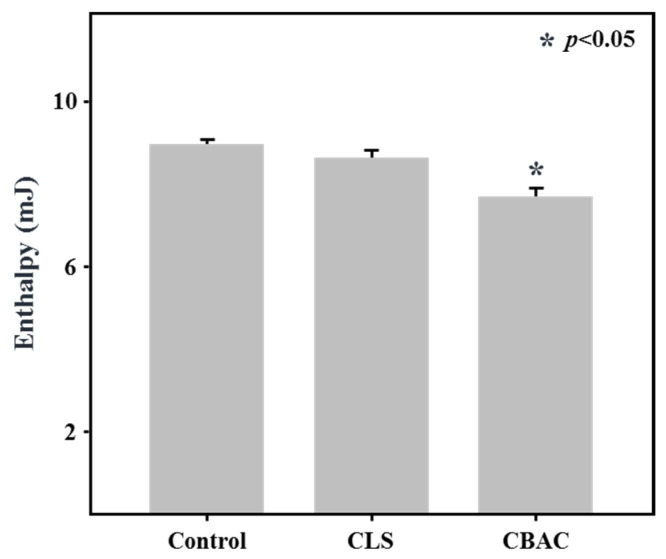
Retrogradation rates of bread samples obtained using differential scanning calorimetry (DSC). After a three-cycle freeze-thaw process, bread dough samples were baked and then stored for 7 days at 4 °C. Asterisks (*) indicate a significant difference at *p* < 0.05 following the Kruskal–Wallis H test.

**Figure 5 foods-10-02269-f005:**
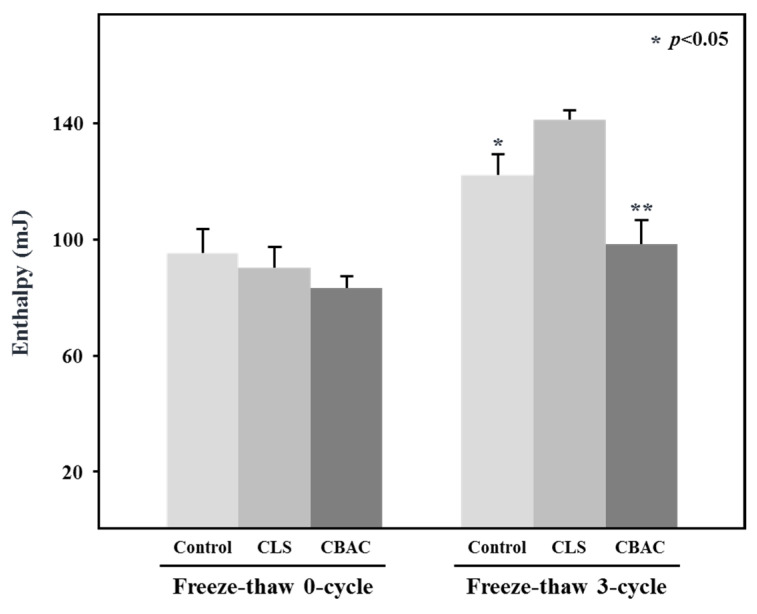
Enthalpy of the freezable water in frozen doughs containing different starches. The dough treated with CBAC had the lowest freezable water enthalpy after a three-cycle freeze thaw process. Asterisk (* or **) indicates a significant difference at *p* < 0.05 following the Kruskal–Wallis H test.

**Table 1 foods-10-02269-t001:** Solubility of the modified starches.

Starch Sample	Water Solubility (mg/mL)	Relative Solubility
Corn starch	12.4 ± 0.00 ^a,b^	1
CBAC-0.5 h ^c^	57.6 ± 0.24	4.65
CBAC-1 h	65.5 ± 0.35	5.28
CBAC-3 h	97.6 ± 0.43	7.87
CBAC-5 h	89.4 ± 0.16	7.21
CBAC-7 h	83.4 ± 0.28	6.73

^a^ Water solubility of corn starch, as reported previously [9]. ^b^ Values are presented as mean ± standard deviation (*n* = 3). ^c^ The CBAC sample was named in accordance with the enzyme reaction time.

**Table 2 foods-10-02269-t002:** Textural properties of bread after storage.

	Textural Properties			
Bread ^a^	Hardness (N)	Springiness	Gumminess (N)	Chewiness (J)
Control	2.52 ± 0.03	1.01 ± 0.00	1.22 ± 0.08	1.23 ± 0.07
CLS	2.30 ± 0.07	1.01 ± 0.00	0.83 ± 0.09	0.83 ± 0.07
CBAC	1.59 ± 0.07	1.00 ± 0.00	0.77 ± 0.03	0.77 ± 0.02

Bread loaves were prepared using frozen dough that had been frozen and thawed three times. Values are presented as mean ± standard deviation (*n* = 3). ^a^ Breads baked with the different starches were stored for 7 days at 4 °C.

## Data Availability

Not applicable.

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
