# Peer review of "Development of Freeze-Thaw Stable Starch through Enzymatic Modification"

_foods, 2021, doi:10.3390/foods10102269_

Round 1
Reviewer 1 Report
Title: Development of freeze-thaw stable starch through enzymatic modification.
Relevant comments:
The research topic is interesting. In the current study, the effects of enzymatic modified starch to increase the quality during the freeze storage. The Introduction are well done however the aim of work is unclear. Material and method are not clear. The interpretations of the results are appropriate however, more discussions and clarifications would be needed. The bibliography used in the discussion supports the explanation of the results obtained. The conclusions are very compressed.
The author must include the following considerations:
Introduction:
Major suggestion
The aim should be increased. What is your hypothesis? What kind of starch modification is more suitable? Reason? What is the importance of investigated parameters?
Minor suggestion
Line 37 Numerous studies is inappropriate if referred only two work (2,11)
Line 74 “functional starch” is not clear. Explain what is meant by this term
In material and methods:
Major suggestion
Usually bread it was made with common wheat why the authors use corn starch?
The dough and bread preparation should be revised. In details the bread preparation is not adequately reported. Regarding the dough I think that the water is essential but is not reported as ingredient. In addition is not clear the used flour and the centesimal composition. All the doughs have reached 500 degrees bra bender? Were used for all doughs the same amount of water?
Why the authors analysed bread stored for 7 day at 4 °C. Usually bread had short shelf life and was stored at room temperature
ANOVA and duncans Statistical analyses were inappropriate. You should use non parametric test to evaluate a small samples.
Minor suggestion
Line 85 add Commercial clean label starch description or references
Line 128-129 clarify the method previously described
Line 157 add all other materials (what and quantity)
Line 164 “dafter” instead of “after”.
Line 180 Why the bread staling was evaluated only after 7 days?
In result:
Major suggestion
Why the results obtained with CLS are not always reported? I suggest to add
Minor suggestion
Table 1 what is a,b in first row of second column?
Line 222 define starch if is a corn starch, or what
Line 231 define what kind of enzymatic modified starch you referred
Line 266-268 This sentence is more appropriate in discussion
Line 267 “other studies” add references
In discussion:
Major suggestion
The author should add a comment for CLS textural analyses. In addition I suppose that different water amount during dough formulation was added. In this case how the different quantity of water added influences the shelf life and the characteristics of the mixture after freezing?
The conclusion should be reported in a separate paragraph. In addition I suggest to increase adding future prospective of this work
Minor suggestion
Line 304 explain how the CBAC improved the resilience of starch and what is means starch resilience
Author Response
Dear Reviewer,
As reviewers’ recommendation, this manuscript was modified in order to answer the reviewers’ questions more clearly.
We would be glad if our manuscript would give you complete satisfaction.
Thank you in advance for your time and consideration.
Best Regards,
Reviewer 2 Report
Quite interesting job. Before deciding to publish it requires the necessary corrections. I am puzzled by two issues raised in the introduction. 1. is frozen dough a good way to healthy, fragrant and desired by consumers bread? 2 on the basis of what the authors say that the bread obtained in this way will meet the requirements of the clen label?
detailed remarks below:
materials and methods 2.1. Chemicals and reagents..according to the publisher's instructions "Materials and Methods: They should be described with sufficient detail to allow others to replicate and build on published results. New methods and protocols should be described in detail while well-established methods can be briefly described and appropriately cited. Give the name and version of any software used and make clear whether computer code used is available. Include any pre-registration codes".... so I do not see any sense in placing this chapter and a rather sketchy description of the materials and reagents used. I think that this content can be added in parentheses when describing the modifications or determinations of selected properties
line 108 "Starch solution (2.5%, w/v) was prepared in distilled water." -From a chemical point of view, starch does not form solutions but a suspension, which sediment after some time. In the introduction, the authors mentioned weak or negligible solubility of starch and this is the reason for the formation of solutions
line 110 "After boiling for 1 h.."-from a chemical point of view, we heat it at a temperature close to 100 Celsius ... we cook soup, sauce
line 114 starch solution (2 mg/mL)- it is safer to sample the modified starch, because basically the starch was heated, so it forms a gruel faster than a solution, but we do not fully know what changes occurred in the presence of chemical reagents
line 144 "..were boiled for 5 min. The samples were then cooled to room temperature and" -please provide the approximate temperature level in brackets. the term room temperature is very confusing, I have 30 degrees C in the summer room and 18 degrees C in the winter,
line 155 "so much " other"
"no other ingredients were added.","All other rials were" - either exactly what was added or without this definition adds nothing,
line 158 "the mixture was kneaded "- I think the ingredients for the bread dough
line 161-162- "After the three freeze-thaw cycles, the dough was fermented again 161 for 100 min and the bread was baked for 40 min." what times, what temperature?
line 171 " after 7 days of storage at 4°C." what level moisture
line 214 "water solubility of enzyme-modified starches are shown"- this is the term water solubility that should be used in the methodology header 2.5.
table 1 - why the statistics are made only for corn starch and there is no information about the remaining significance of time effect on water solubility
I assume it's due to negligence but the authors forgot to summarize the results of their work no conclusions !!!!
Author Response

(The authors gave the same response as above.)

Round 2
Reviewer 1 Report
I thank the authors for taking all comments into consideration. However,no details on the comments actually considered were given in the attached
reviewer's response. In some cases it would seem that the suggestion has not
been taken into consideration:
In particular no substantial modification was reported in text for:
-The aim should be increased. What is your hypothesis? What kind of starch
modification is more suitable? Reason? What is the importance of
investigated parameters?
- add the centesimal composition of flour and if the same for all experiment.
All the doughs have reached 500 degrees bra bender?
Were used for all doughs the same amount of water?
-Why the authors analysed bread stored for 7 day at 4 °C.
Usually bread had short shelf life and was stored at room temperature
-The author should add a comment for CLS textural analyses. In addition
I suppose that different water amount during dough formulation was added.
In this case how the different quantity of water added influences the shelf
life and the characteristics of the mixture after freezing?
Author Response
We really appreciate for your insightful comments. Your recommendations were very helpful and extended our point of view regarding bread analysis study.
Q1. The aim should be increased. What is your hypothesis?
- There are many researches which explain that the addition of materials which bind water molecules improves the quality of frozen dough (Introduction, line 51-58).
We hypothesized that the form of amylopectin clusters with many short branches had an excellent water holding capacity, therefore, when it was used for frozen dough, its freeze-thaw stability will be high [1].
As reviewer's recommendation, our hypothesis and aim of study was re-written in Introduction part (line 77-80).
Q1-1. What kind of starch modification is more suitable? Reason? What is the importance of investigated parameters?
- The suitable starch should keep the water molecules in dough tightly in drastic condition. Usually, the more short branches a starch has, the more effectively it prevents syneresis [2]. Therefore, the change of size chain length distribution of CBAC was investigated mainly.
Q2. Were used for all doughs the same amount of water?
- In every sample, the same amount of water (220 mL) was added (line 151). We corrected the method more clearly. Our mistake, ‘w/v’ was corrected to ‘w/w’ (line 153-154).
Q3. Add the centesimal composition of flour and if the same for all experiment
- We totally agree with you that the composition of flour could be changed ‘slightly’ during sample preparation. To confirm that this slight difference is negligible value or not, pretest had been done without freeze-thaw process first. In the pre-test, the sample groups showed similar qualities during baking in ‘Normal’ condition (bread loaf volume, freezable water, appearance etc.), you might comprehend that if you see Fig. 3. Even after freeze-thaw process, there was no significant difference in appearance (bread loaf volume) between samples.
We strongly believe that the slight difference in flour composition was not adjustable exactly and did not make an effect on the freeze-thaw stability test.
Q4. All the doughs have reached 500 degrees bra bender?
- Unfortunately, we could not check bra bender unit due to the lack of proper analytical system in our research group. For preparation of dough and bread, bread maker (Cuchen, Seoul, Korea) was employed.
Even though we do not have the information, all of our samples were followed the manufacturer’s instruction manual and recipe properly at the same condition, which could satisfy the quality of dough for the baking.
Q5. Why the authors analyzed bread stored for 7 day at 4 °C. Usually bread had short shelf life and was stored at room temperature.
- There are many factors which are related with shelf life of bread. Among them, we focused on ‘retrogradation of starch’ in bread because CBAC (modified starch additive) is strongly related with starch-water molecule interaction. Therefore, accelerated storage test was performed for 7-day at 4°C, which is usual condition of retrogradation test for starchy food [3, 4].
Q6. Should add a comment for CLS textural analyses. I suppose that different water amount during dough formulation was added. In this case, how the different quantity of water added influences the shelf life and the characteristics of the mixture after freezing?
- We are sorry that our typo made you confused. In every sample, the same amount of water (220 mL) was added (line 151). The comment for CLS was added previously in line 251-253.
- As shown in figure 5, different quantity of water occurred during 3-cycles of freeze-thaw process. Every group of dough sample lost water molecules during freeze-thaw. However, the dough treated with CBAC had the lowest freezable water enthalpy, which means CBAC treated dough released less amount of water than other groups.
- We agree that water content in dough is one of the most important point in texture and shelf-life in bread. The previous research reported that bread retaining much more amount water tends to improve the shelf-life by reducing the increase in firmness over time [5]. The influence of water quantity was also added in discussion part (line 311-312).
REFERENCES
[1] Mukerjea, R.; Robyt, J.F. Isolation, structure, and characterization of the putative soluble amyloses from potato, wheat, and rice starches. Carbohydr Res 2010, 345, 449-451.
[2] Jobling, S.A.; Westcott, R.J.; Tayal, A.; Jeffcoat, R.; Schwall, G.P. Production of a freeze-thaw-stable potato starch by antisense inhibition of three starch synthase genes. Nat Biotechnol 2002, 20, 295-299.
[3] J-H Shim, N-S Seo, S-A Roh, J-W Kim, H Cha, K-H Park, Improved bread-baking process using Saccharomyces cerevisiae displayed with engineered cyclodextrin glucanotransferase. J Agric Food Chem. 2007 55(12):4735-40.
[4] S-H Woo, Y-J Shin, H-M Jeong, J-S Kim, D-S Ko, J-S Hong, H-D Choi, J-H Shim, Effects of maltogenic amylase from Lactobacillus plantarum on retrogradation of bread, J Cereal Sci 2020 93: 102976.
[5] Gil, M.J.; Callejo, M.J.; Rodríguez, G. Effect of water content and storage time on white pan bread quality: instrumental evaluation. Z Lebensm Unters F A 1997, 205, 268-273.

Reviewer 2 Report
The work shows a thorough improvement and the corrections made significantly increase its quality. I still suggest:
line 92 - native starch does not form solutions, but suspensions
The work can be published in this form.
Author Response
We are grateful for your insightful comments. Your recommendations were very helpful and extended our point of view.
Q1. native starch does not form solutions, but suspensions
- As reviewer's comment, the 'solution' was changed to 'suspension'.
